# IBD: An Interpretable Backdoor-Detection Method via Multivariate Interactions

**DOI:** 10.3390/s22228697

**Published:** 2022-11-10

**Authors:** Yixiao Xu, Xiaolei Liu, Kangyi Ding, Bangzhou Xin

**Affiliations:** Institute of Computer Application, China Academy of Engineering Physics, Mianyang 621900, China

**Keywords:** deep neural network, backdoor detection, interpretable deep learning

## Abstract

Recent work has shown that deep neural networks are vulnerable to backdoor attacks. In comparison with the success of backdoor-attack methods, existing backdoor-defense methods face a lack of theoretical foundations and interpretable solutions. Most defense methods are based on experience with the characteristics of previous attacks, but fail to defend against new attacks. In this paper, we propose IBD, an interpretable backdoor-detection method via multivariate interactions. Using information theory techniques, IBD reveals how the backdoor works from the perspective of multivariate interactions of features. Based on the interpretable theorem, IBD enables defenders to detect backdoor models and poisoned examples without introducing additional information about the specific attack method. Experiments on widely used datasets and models show that IBD achieves a 78% increase in average in detection accuracy and an order-of-magnitude reduction in time cost compared with existing backdoor-detection methods.

## 1. Introduction

With the application of deep-learning techniques in computer vision, services based on image and video classification (e.g., autonomous driving [1,2,3], face recognition [4,5]) have become a significant convenience in our daily lives. However, security incidents caused by deep-learning model crashes occur on a regular basis. As a result, the vulnerability of deep-learning algorithms has become a major public concern, as well as the focus of current research. Recent research indicates that deep neural networks (DNNs) are vulnerable to backdoor attacks in which adversaries inject backdoors into victim models via data poisoning [6] or structure modifying [7]. The infected model behaves normally on benign examples, while maliciously when handling examples with specific triggers.

Sensors are more and more implicated with deep-learning techniques due to the progress in the internet of things [8,9,10]. However, existing studies on backdoor attacks against DNNs have demonstrated that backdoor attacks may occur throughout the lifetime of a deep-learning model, including model during training [6,8,9], model transmission [7], and model application [10], which makes them a great threat against deep-learning-based sensors. Meanwhile, attackers may maliciously use the sensors and inject a backdoor [11]. Therefore, mitigating the threat of backdoor attacks is critical. Wang et al. [12] first proposed to detect backdoor models by reversing potential triggers. Some subsequent works focused on the characteristics of triggers and proposed to detect backdoor models via image mixture [13] or adversarial example generation [14].

However, despite the initial success of existing backdoor-defense methods against already-known attacks, their transferability and interpretability are still not guaranteed. Several of the latest backdoor attack approaches [15,16] have shown the ability to bypass existing defense methods, as the triggers of these new attacks exhibit different characteristics from previous attacks. In particular, existing backdoor-defense methods are still faced with the following challenges:

(**Challenge 1**) Due to the lack of theoretical analysis from the perspective of interpretable methods, existing backdoor-defense methods are mainly empirical. Thus, the effectiveness of these methods is not guaranteed.

(**Challenge 2**) Most existing defense methods are based on the summation of the characteristics of previous backdoor attacks, which makes them less effective when handling unknown new attacks, especially when new attacks are designed to have different characteristics from previous methods.

(**Challenge 3**) Methods based on trigger reversing or model training introduce additional significant computational costs. With the increase in model and dataset size, the computational resources required may become unacceptable.

Therefore, it is necessary to investigate the rationale of backdoor attacks using interpretability methods and further propose transferable and robust defense approaches.

**Interpretable backdoor-detection methods.** To address the above challenges, we propose **IBD**, an interpretable backdoor-detection method via multivariate interactions. We first analyze the multivariate interactions of different categories in classification models and demonstrate that the abnormal cooperative relationships in poisoned models can be used to detect backdoor models. Furthermore, we propose a multivariate-interaction-based approach for backdoor model detection, which enables defenders to perform detection under black-box constraints with acceptable time costs. Figure 1 gives a visualization of the overall structure of IBD. Our main contributions can be summarized as follows:Taking advantage of information theory techniques, we theoretically analyze the multivariate interactions of features from different categories and reveal how the backdoor works in infected models (**addressing Challenge 1**);Following the theoretical analysis, we propose a theoretical-guaranteed defense method IBD, an interpretable backdoor-detection method via multivariate interactions. IBD outperforms existing defense methods for detecting both baseline attacks (attacks that existing defense methods have taken into consideration) and new attacks (attacks that can bypass existing defense methods) (**addressing Challenge 2**);Guided by the theoretical foundations of IBD, we further accelerate IBD by simply calculating the summation of the logits of several input examples from different categories. Therefore, defenders can detect malicious models without additional information about the parameters and structures of the target model (**addressing Challenge 3**).

We evaluated our multivariate-interaction-based backdoor-detection method and the simplified version on different datasets. Experimental results show that the multivariate-interaction-based method can divide benign and poisoned models with high detection accuracy. Furthermore, IBD achieves a 99% reduction in time cost and a 78% increase in detection accuracy on average compared with existing defense methods.

## 2. Related Work

### 2.1. Deep Backdoor Attack

The deep backdoor attack, also known as the Trojan attack, is originally a special variant of the model poisoning attack. Adversaries inject the backdoor into deep-learning models by poisoning the training dataset with examples containing a specific trigger [6]. BadNets [6] was the first backdoor-attack method against deep neural networks, which gave the definition and explored the fundamental approaches of backdoor attacks. However, poisoned examples with visible triggers and incorrect labels are easy to detect by humans or autonomous detection algorithms. Therefore, later studies on backdoor attacks mainly focus on the invisibility of triggers and the label-consistency of poisoned examples. Chen et al. [17] proposed a blending-based backdoor-attack method that blends instead of attaching triggers to poisoned examples. Nguyen et al. [15] proposed WaNet, an elastic-warping-based backdoor-attack method using warping-based triggers that are stealthier to human inspection. Doan et al. [16] proposed LIRA, a learnable, imperceptible, and robust backdoor-attack method. LIRA formulates the backdoor attack process in a constrained problem of finding the optimal trigger generator and the optimal victim classifier. It alternately updates the generator and the classifier and, hence, makes the triggers sample-specific and invisible.

Several other studies have explored backdoor attacks that occur throughout different stages [7,18]. Tang et al. [7] proposed a training-free backdoor-attack method by simply inserting a tiny Trojan model (TrojanNet) into the target model. Thus, the backdoor can be injected into the victim model during the transmission process of models. Rakin et al. [18] took advantage of the bit-flip method [19] and proposed a test-time backdoor-attack method: TBT. TBT injects a backdoor into a victim model by directly modifying the model parameters stored in the main memory (DRAMs). Therefore, TBT can bypass most training-time- or fine-tune-based defense methods since it occurs in test time.

According to the above research, backdoor attacks can occur during the whole lifetime of deep-learning models, including the training, transmission, and deployment processes. Furthermore, the format of triggers is flexible, which further increases the difficulty of backdoor defense.

### 2.2. Deep Backdoor Defense

To alleviate the increasing threat of deep backdoor attacks, several backdoor-defense methods are proposed. Li et al. [20] divided existing defense methods into two categories based on their theoretical foundations: empirical and certified backdoor defenses. Empirical methods extract some features of existing attacks and achieve decent performance when dealing with already-known attacks. Neural Cleanse [12] is the first backdoor-defense method that detects backdoor models by reversing potential triggers. However, the assumption of Neural Cleanse that backdoor triggers are smaller than benign features is not transferable enough, i.e., triggers may have different features that Neural Cleanse does not take into consideration. Gao et al. [13] proposed STRIP, which detects poisoned examples by performing an image linear blend. The prediction of the blended images trends to stay constant when any test image contains a specific trigger. Some later researchers attempted to use deep-learning models to detect backdoors, Kolouri et al. [14] took advantage of the fact that backdoor models may behave differently from clean ones when dealing with adversarial examples and proposed universal litmus patterns (ULPs). However, ULPs require generating universal adversarial patterns for each category of the target model, which will cause an unacceptable increase in time cost with the increase in the number of categories.

In this paper, we focus on the detection of poisoned models and examples. Different from most previous defense methods that are based on prior assumptions about existing attacks, we initially explore the training process of victim models and the functioning principle of backdoors from the perspective of interpretable methods. Then, we use the principles to guide backdoor detection without prior knowledge about the training dataset or the type of trigger. Therefore, our method achieves better transferability against different attack methods.

### 2.3. Shapley Value

Some previous studies on interpretable theorems of deep-learning algorithms formulate the classification problem as an n-player game from the perspective of game theory [21,22,23]. The fundamental theorem of these studies is the Shapley value [24], which is considered as a fair algorithm to measure the contributions of different players in a game [25]. Consider an n-player game $v:S_{i}\to R$, where N={1,2,⋯,n} denotes the set of all players and $S=\{S_{1},S_{2},\cdots ,S_{2^{n}}\}$ denotes the superset made up by all $2^{n}$ subsets of *N*. According to the definition of the Shapley value [24], the unbiased contribution of the *i*-th player can be calculated as:(1)$$\phi_{v}\left(i|N\right)=\sum_{S\subseteq N}\frac{(n-|S|-1)!|S|!}{n!}\left[v\left(S\cup \left\{i\right\}\right)-v\left(S\right)\right]$$
where $\frac{(n-|S|-1)!|S|!}{n!}$ denotes the probability of a certain subset $S_{i}$ and [v(S∪{i})−v(S)] denotes the variance of the reward before and after the introduction of the *i*-th player.

Later, Zhang et al. [23] proposed to use the Shapley value to interpret the multivariate interactions of different parts of the input example during the classification process of deep neural networks. Given two players *i* and *j* with their Shapley values $\phi_{v}\left(i|N\right)$ and $\phi_{v}\left(j|N\right)$, the interaction between *i* and *j* is defined as:(2)$$B\left(S_{ij}\right)=\phi \left(S_{ij}|N^{{}^{\prime }}\right)-\left[\phi \left(i|N_{i}\right)+\phi \left(j|N_{j}\right)\right]$$
where $B(S_{ij})$ represents the interaction of the player *i* and the player *j*, and $\phi (S_{ij}|N^{{}^{\prime }})$ denotes the Shapley value of the player ij when considering *i* and *j* as a unit.

## 3. Methodology

### 3.1. Problem Formulation

**Multi-category classification.** The goal of the standard supervised classification task is to learn a parameterized mapping function $F_{\theta }:X\to C$, where X is the input domain and C denotes the set of target classes [16]. Generally, the classification process contains two stages: feature extraction, and prediction. The mapping function $F_{\theta }$ extracts *n* features $\left\{f_{1},f_{2},\cdots ,f_{n}\right\}$ from the input example *x* and then makes predictions based on these features. Therefore, the classification task can also be formulated as an n-player game from the perspective of game theory [23]. In this game, *n* players denote *n* features of the input example, and the reward denotes the output scalar of the prediction function. All players cooperate and compete with each other to obtain the overall reward. In this paper, we focus on the image classification task, where *n* features are implemented as the neurons in the layer before the last full-connection layer of the image classification model; the reward is implemented as the score of the true category before the softmax layer.

**Deep backdoor attack.** Following the definition of deep backdoor attacks [15,16], the adversary designs a transformation function *T* to transform a clean example (x,y) into a poisoned example $\left(\hat{x}=T\left(x\right),\hat{y}\right)$, where *y* and $\hat{y}$ denote the clean label and the target label determined by the adversary, respectively. Then, the adversary injects the backdoor into the victim model *F* by poisoning the training dataset [15,16] or by modifying the model structure [7] or parameters [18] directly. Therefore, the behavior of *F* is altered as: $F\left(x\right)=y,F\left(T\left(x\right)\right)=\hat{y}$.

**Deep backdoor detection.** For the detection of deep backdoor attacks, the goals can be divided into model-based detection and sample-based detection. Model-based detection methods attempt to classify poisoned models and benign models. Sample-based detection methods aim to detect triggers in poisoned examples and alleviate the risk by removing triggers or performing transformations on poisoned examples.

In this paper, we are interested in detecting backdoor attacks in image classification models. The main focus of our method is to propose a backdoor model detector $D_{m}$ together with a poisoned example detector $D_{s}$ for model-based detection and sample-based detection, respectively. The design principles of the two detectors are based on the interpretable theorem of DNNs. Specifically, for a k-category image classification model *F*, we use the backdoor model detector $D_{m}$ and a small set of images $S=\{x_{1},x_{2},\cdots ,x_{m}\}$ from *k* classes to evaluate the inter-class multivariate interactions of image features from different classes: $target=D_{m}\left(F,S\right),target\in \left\{\phi ,0,1,\cdots ,k-1\right\}$. For backdoor models, $D_{m}$ will output the target class $\hat{y}=D_{m}\left(F,S\right)$ for the backdoor. Further, the poisoned example detector $D_{s}$ can determine whether a test example *x* from the target class $\hat{y}$ contains the trigger based on the inner-image multivariate interactions of different features extracted from the test example: $poisoned=D_{s}\left(F,x\right),poisoned\in \left\{0,1\right\}$.

### 3.2. Multivariate Interactions of Benign and Backdoor Models

Consider the training process of a benign k-category image classification model $F_{\theta }$ with parameters θ; the forward process makes predictions based on *n* features $\left\{f_{1},f_{2},\cdots ,f_{n}\right\}$ extracted from the input image *x*. Then, the backward process calculates the gradients of different sets of neurons representing different features and updates the parameters. When focusing on a single feature $f_{i}$ of *x*, the forward–backward training process iteratively adjust the weights $\left\{\alpha_{i}^{1},\alpha_{i}^{2},\cdots ,\alpha_{i}^{k}\right\}$ of $f_{i}$ to *k* different classes. For classes containing $f_{i}$, the weights gradually increase during the training process, while for classes where $f_{i}$ rarely appears, the weights tend to decrease. Finally, a well-trained image classification model contains the appropriate feature information of *k* different classes. For each category, the model holds a set of positive features that contribute most to that category and a set of negative features that deny the prediction of the category. Figure 2 illustrates a simple example of the parameter optimization process of deep-learning models. The model’s parameter is initialized as 0.1 and updated by two input examples from different categories. After the two-step update process, class 0 builds up positive and negative relationships with feature 0 and feature 1, respectively. For two different categories, if they share many positive features, they usually show a tendency to cooperate in the classification results, while categories compete with each other when the positive features of one category tend to be the negative features of the other.

However, when considering the training process of a backdoor image-classification model $F_{\phi }$, the cooperative and competitive relationships between features of different classes may change due to the injection of backdoors. Specifically, for the training process of the target class, $F_{\phi }$ is fed with three different types of features: (1) benign features of benign examples, (2) trigger features of the trigger, and (3) disturbing features of poisoned examples. During the parameter updating process, all of these features build up a positive relationship with the prediction of the target class. As shown in Figure 3, for a certain class, different features have different relationships with the class if the model is benign. If the model is poisoned and the class is the target class of the backdoor attack, then different features have similar relationships (positive) with the target class.

To quantify the relationships of features in benign models and backdoor models and theoretically analyze the basic characteristics of backdoors of deep neural networks, we introduce the Shapley value and multivariate-interaction evaluation strategy. According to Equation (Equation 2), Zhang et al. [23] defined the increment of the Shapley value as a quantization relation of two different features of an input example. In the study region of our interests, we consider sets of features of different categories. Therefore, we reconstruct the set of input features and consider the classification process as a two-player game. Specifically, for a k-category image classification model, *F*, $\left(x_{1},k_{1}\right)$ and $\left(x_{2},k_{2}\right)$ are two input examples from two categories $k_{1}$ and $k_{2}$, respectively. The feature extraction function *f* of model *F* embeds an input example into *n* features. Therefore, the embedded features of $x_{1}$ and $x_{2}$ can be represented as $f\left(x_{1}\right)=\left\{f_{1}^{1},f_{1}^{2},\cdots ,f_{1}^{n}\right\}$, $f\left(x_{2}\right)=\left\{f_{2}^{1},f_{2}^{2},\cdots ,f_{2}^{n}\right\}$. We consider these features as two subsets of the feature set of $k_{1}$ and $k_{2}$; since our goal is to evaluate the relationships between two categories, we reconstruct the input feature set with a union of the subsets of $f(x_{1})$ and $f(x_{2})$. Let $s_{1}^{1}=\left\{f_{1}^{1},f_{1}^{2},\cdots ,f_{1}^{\frac{n}{2}}\right\}$, $s_{1}^{2}=\left\{f_{1}^{\frac{n}{2}+1},f_{1}^{\frac{n}{2}+2},\cdots ,f_{1}^{n}\right\}$; then, $f(x_{1})$ can be represented as $f\left(x_{1}\right)=\left\{s_{1}^{1},s_{1}^{2}\right\}$. Similarly, $f\left(x_{2}\right)=\left\{s_{2}^{1},s_{2}^{2}\right\}$. Thus, we can reconstruct two input feature sets $S_{1}=\left\{s_{1}^{1},s_{2}^{2}\right\}$ and $S_{2}=\left\{s_{1}^{2},s_{2}^{1}\right\}$ where each set contains features from different categories. According to Equation (Equation 1), we define the reward *v* of the game as the output logits for the target class $k_{1}$; we can calculate the Shapley value $\phi_{v}\left(s_{1}^{1}|S_{1}\right)$, $\phi_{v}\left(s_{2}^{2}|S_{1}\right)$, $\phi_{v}\left(s_{1}^{2}|S_{2}\right)$, and $\phi_{v}\left(s_{2}^{1}|S_{2}\right)$. Then, we define the relationship of features from two categories:(3)$$B\left(S_{k_{1}},S_{k_{2}}\right)=\sum \phi \left(s|S^{{}^{\prime }}\right)-\left[\phi \left(i|S\right)+\phi \left(j|S\right)\right]$$
when $B(S_{k_{1}},S_{k_{2}})>0$ features of $k_{1}$ and $k_{2}$ tend to cooperate with each other, while $B(S_{k_{1}},S_{k_{2}})<0$ illustrates that features of $k_{1}$ and $k_{2}$ tend to compete with each other. Given a k-category classification model and Equation (Equation 3), we can calculate a k×k matrix *M* to represent the relationships of different categories’ features. The *ij*-th value in *M* quantifies the relationship between the features of the *i*-th category and the features of the *j*-th category. For example, Figure 4 gives a visualization of the relationship matrix of a 10-category benign classification model trained on CIFAR-10. As shown in the image, features of one category compete and cooperate with features from other categories and play together to obtain the final prediction result.

### 3.3. A Simple Approach for Backdoor Model Detection

The differences in multivariate interactions of benign and backdoor models enable us to detect backdoor models using the multivariate-interaction matrix. Figure 5 gives an example of the relationship matrix of backdoor models poisoned by the BadNet [6] attack method. The model is trained on CIFAR-10 and the target class for the attack is set to be 1. As shown in Figure 5, the relationships between features of category 1 with others show a different trend from those in benign models (as shown in Figure 4). Therefore, we can use the sum of each column of the relationship matrix as a feature to distinguish benign and backdoor models. Given a boundary value *b*, the k×k multivariate-interaction matrix *M* of a *k*-category test model *F*, the backdoor risk values $\left\{r_{1},r_{2},\cdots ,r_{k}\right\}$ can be calculated as $r_{i}=\sum_{j=1}^{k}M_{ij}$, if $r_{i}>b$; then, the test model is poisoned and the target attack class is class *i*.

However, extracting features from input examples requires access to intermediate layers of the test model, which is often impracticable in real-world conditions. To achieve backdoor detection under black-box constraints where only the prediction result (logits) of the test model is available, we propose a simple approach for the backdoor model detection method exploiting the idea of multivariate interactions. Specifically, for a *k*-category classification model *F*, we require the defender to collect *k* small sets of *n* test examples $\left\{\left\{x_{1}^{1},x_{1}^{2},\cdots ,x_{1}^{n}\right\},\left\{x_{2}^{1},x_{2}^{2},\cdots ,x_{2}^{n}\right\},\cdots ,\left\{x_{k}^{1},x_{k}^{2},\cdots ,x_{k}^{n}\right\}\right\}$ for each category (e.g., n=3 means collecting 3 examples for each category); we do not have any assumptions about these test examples (they could even contain a trigger). Then, the defender feeds these test examples into the test model and calculates the sum of their prediction logits as the risk vector:(4)$$r=\left(\begin{array}{c} r_{1}\\ r_{2}\\ \cdots \\ r_{k} \end{array}\right)=\sum_{i}\sum_{j}F\left(x_{i}^{j}\right)$$

Given a risk boundary *b*, if the *i*-th component $r_{i}$ of the risk vector r is larger than *b*, this model is considered to be a backdoor model and the *i*-th category is the target attack class.

Our proposed backdoor model detection method is based on the idea of multivariate interactions and is a macroscopic representation of the interaction of different categories of features on the classification results. Compared with existing backdoor model detection methods, our method is more effective and efficient, with the following two characteristics:**Transferable.** Most existing backdoor-detection methods have some assumptions about potential attack methods (e.g., Neural Cleanse [12] requires the trigger to be smaller than the natural features). Attackers can change the form of triggers guided by these assumptions, which makes existing detection methods non-transferable to new attacks. IBD does not require defenders to have prior knowledge about the potential attacks. Therefore, it is transferable to detect different types of backdoor attacks, including some state-of-the-art attack methods that can bypass most existing defense methods [15]. Meanwhile, it is difficult for attackers to manipulate the multivariate interactions of features in the target model; thus, our methods are still robust against defense-known attacks;**Cost-Friendly.** Existing backdoor-detection methods often require defenders to collect a set of reliable test examples or to perform thousands of forward–backward steps on the target model, which leads to both high preparation costs and time costs. Compared with existing methods, our method is cost-friendly, since it only requires several forward processes of the classification model and does not require the test examples to be trustful (without triggers).

## 4. Experimental Results

### 4.1. Defense Setting

**Datasets.** Following most previous backdoor-attack methods [6,15,16,17], we used MNIST [26], CIFAR-10 [27], and GTSRB [28] to evaluate our method. MNIST is a handwritten digits dataset and contains 60,000 training examples and 10,000 examples written by 250 different people. CIFAR-10 is a labeled subset of the 80 million tiny images dataset, which contains 10 categories with 60,000 images. Each example in CIFAR-10 is sized to 32×32 and has three color channels. GTSRB is a German traffic sign recognition dataset containing 43 categories with 39,209 training examples and 12,630 examples. Each example in GTSRB is sized to 32×32 and has three color channels.

**Victim Models.** Following BackdoorBox [20], we used two different image recognition models as victim models when performing backdoor attacks on different datasets. For MNIST, we used the baseline network for MNIST proposed in BadNet [6], which consists of two convolutional layers and two fully connected layers. As for CIFAR-10 and GTSRB, we used ResNet-18 as the victim model.

**Backdoor-attack methods.** Among the above-mentioned backdoor-attack methods in related works, we considered three methods as target methods for the following reasons. Our first target method is BadNet [6], which was the first backdoor-attack method. BadNet is widely used to evaluate the performances of different backdoor-detection methods; thus, we consider it as the baseline backdoor-attack method. Then, we further introduce two of the latest backdoor-attack methods: WaNet [15] and LIRA [16]. WaNet takes advantage of image-warping technologies and designs warping-based invisible triggers. According to the experiment result [15], WaNet can bypass most existing backdoor-defense methods, including Neural Cleanse [12], Fine-Pruning [29], STRIP [13], and GradCam [30]. LIRA, different from most other backdoor-attack methods, attempts to use neural networks to generate sample-specific and invisible triggers. Similar to WaNet, the experimental results of LIRA also prove that it is robust against most existing defense methods [16]. Therefore, we use WaNet and LIRA as defense-robust attack methods to evaluate the effectiveness of our method.

**Baseline Defense Methods.** We compared our method with Neural Cleanse [12], which detects backdoor models by reversing potential triggers.

**Metrics.** For backdoor model detection, we mainly focused on the detection accuracy of our method, which consists of the true positive rate (TPR) and the true negative rate (TNR). Meanwhile, we compared the computational cost of different methods.

**Detail Settings.** According to the backdoor-model-detection process, we randomly select 10 examples for each category and sum up their output logits. Since IBD does not require these test examples to be trustworthy, the test dataset can be easily obtained.

### 4.2. Evaluate Interpretable Method

First, we evaluate our multivariate-interaction-based interpretable method on three different datasets against three attack methods. Following the theoretical analysis of multivariate interactions, we calculate the mean relationship values of benign and poisoned models using Equation (Equation 3). According to Table 1, for benign classes, the mean relationship values of benign and poisoned models are similar, while the mean relationship values of the target classes are much larger. The experimental result demonstrates that features of different categories show an abnormally cooperative relationship in the poisoned models, which can be used to detect backdoor models.

### 4.3. Performance Comparison

We compared the performance of IBD and Neural Cleanse against three baseline backdoor-attack methods on three different datasets shown in Table 2. Each attack method achieved an attack success rate higher than 98%, which proves the efficiency of these baseline attacks. When constraining the false positive rate to be 0%, i.e., all the benign models are correctly classified, it was observed that IBD achieves a much higher true positive rate than Neural Cleanse (a 78% increase in true positive rate on average), which proves the effectiveness of IBD. Meanwhile, the detection accuracy of IBD on CIFAR10 and GTSRB datasets against three attacks is 100%, which demonstrates the ability of IBD for highly precise environments where the training cost of a model is expensive. Additionally, the detection accuracy of Neural Cleanse against WaNet and LIRA attacks is significantly lower than against BadNet, because the assumptions about triggers are not suitable when handling WaNet and LIRA backdoor models. In contrast, IBD does not have a prior assumption about the backdoor triggers, thus allowing better transferability and robustness in dealing with unknown attacks.

Then, we compared the overall performance of IBD and Neural Cleanse using histograms with RUG shown in Figure 6. According to the figure, we have the following observations. (1) Compared with Neural Cleanse, the detection values of benign and poisoned models calculated by IBD can be significantly distinguished; thus, IBD achieves a high true positive rate (TPR) while keeping a low False Positive Rate (FPR). (2) As referred to in previous work [12,14], our experimental results also prove that Neural Cleanse can effectively detect backdoor models attacked by BadNet. However, Neural Cleanse achieves the best performance when dealing with the MNIST dataset and CIFAR10 dataset, and behaves less effectively for a more complex dataset such as GTSRB. In contrast, IBD keeps a high detection accuracy when dealing with models trained on a different dataset. (3) The latest backdoor-attack methods such as WaNet and LIRA could bypass Neural Cleanse with a high probability, since Neural Cleanse is based on assumptions about the potential triggers and the assumptions are no longer appropriate for new attack methods. IBD does not require additional information about the potential attack method and is based on the multivariate interactions of target models, which cannot be manipulated by attackers, so IBD remains effective against the latest backdoor-attack methods.

We also compared the time cost of IBD and Neural Cleanse detecting one model trained on different datasets shown in Table 3. As analyzed in previous work [14], the time complexity of Neural Cleanse is $O(K^{2})$ since it requires performing targeted adversarial attacks from each category to another. Therefore, with the increase in the complexity of the training dataset, the time cost of Neural Cleanse becomes unacceptable. In contrast, the time complexity of IBD is O(K) because it only requires some forward inference on the target model, which makes IBD a cost-friendly and large-dataset-applicable backdoor-detection method.

## 5. Conclusions

In this paper, we propose IBD, an interpretable backdoor-detection method via multivariate interactions. Using information theory techniques, IBD reveals how the backdoor works from the perspective of multivariate interactions of features. Due to the training process of backdoor models, the features of other categories contribute abnormally to the prediction of the target label. Based on the interpretable theorem, IBD enables defenders to detect backdoor models and poisoned examples without introducing additional information about the specific attack method. Meanwhile, we propose a simple way to detect backdoor models on the basis of our theoretical analysis. By summarizing the output logits of several input examples, IBD significantly reduces the time cost for backdoor detection and enables defenders to perform detection under black-box conditions. Experiments on widely used datasets and models show that IBD achieves a 78% increase on average in detection accuracy and an order-of-magnitude reduction in time cost compared with existing backdoor-detection methods. The most relevant future work is to explore the performance of IBD against detection-known attacks.

## Figures and Tables

**Figure 1 sensors-22-08697-f001:**
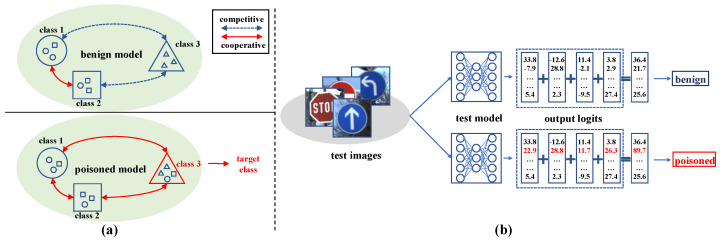
The overall structure of IBD. (**a**) Multivariate interactions of different categories in benign and poisoned models. (**b**) A multivariate-interaction-based simple approach for backdoor model detection.

**Figure 2 sensors-22-08697-f002:**
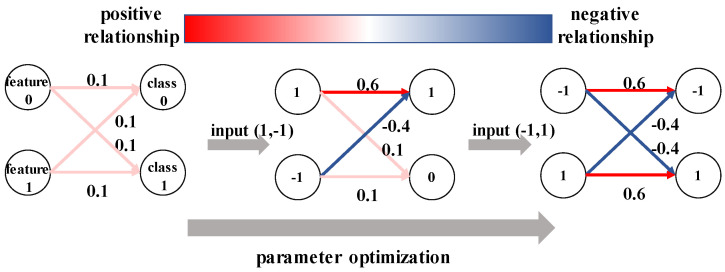
An example of the parameter-optimization process of deep-learning models. After the two-step update process, two output classes build up positive and negative relationships with different features.

**Figure 3 sensors-22-08697-f003:**
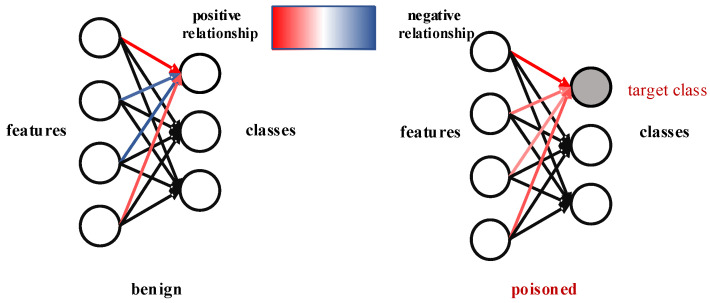
Comparison of the relationships between features and classes of benign models and backdoor models. For benign models, features have both positive and negative relationships with a prediction class, while for backdoor models, most features show a cooperative tendency in the classification of the target class. Features in relation to other classes are marked with black arrows.

**Figure 4 sensors-22-08697-f004:**
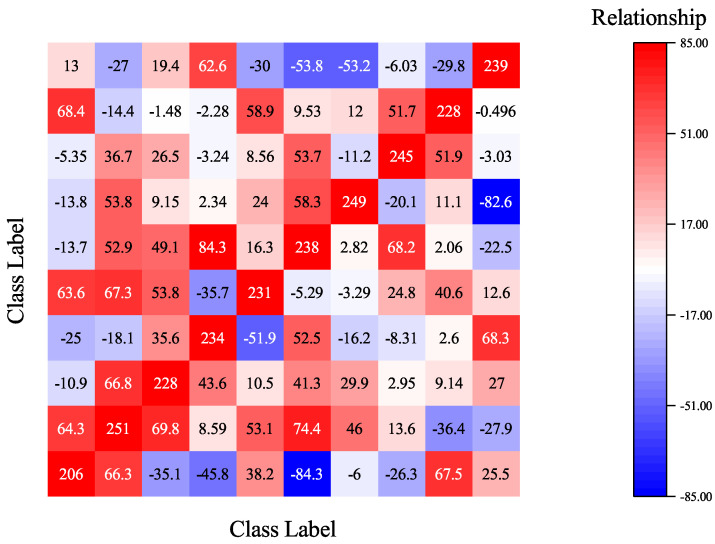
A heat-map visualization of the relationship matrix of a 10-category image classification model. The value in the *ij*-th box quantifies the relationship between the features of the *i*-th category and the features of the *j*-th category, where positive values represent cooperative relationships, while negative values represent competitive relationships.

**Figure 5 sensors-22-08697-f005:**
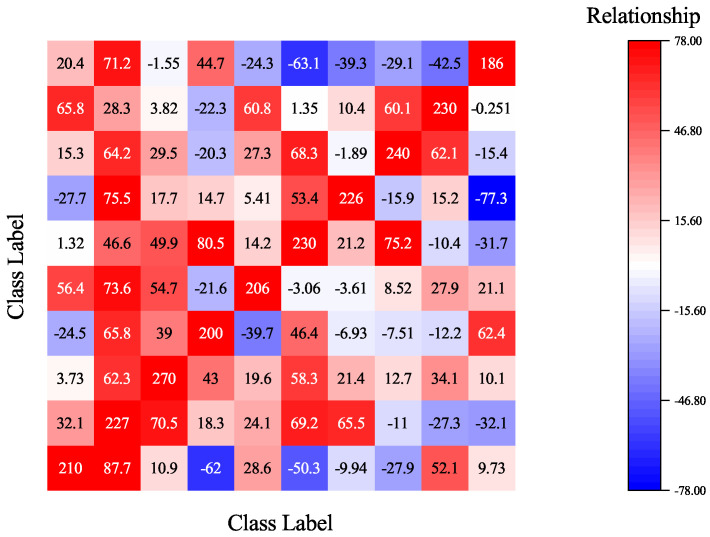
A heat-map visualization of the relationship matrix of a backdoor model poisoned by the BadNet attack method. The target attack class is class 1. As shown in the figure, the features of class 1 show an abnormal cooperative relationship with features of other categories.

**Figure 6 sensors-22-08697-f006:**
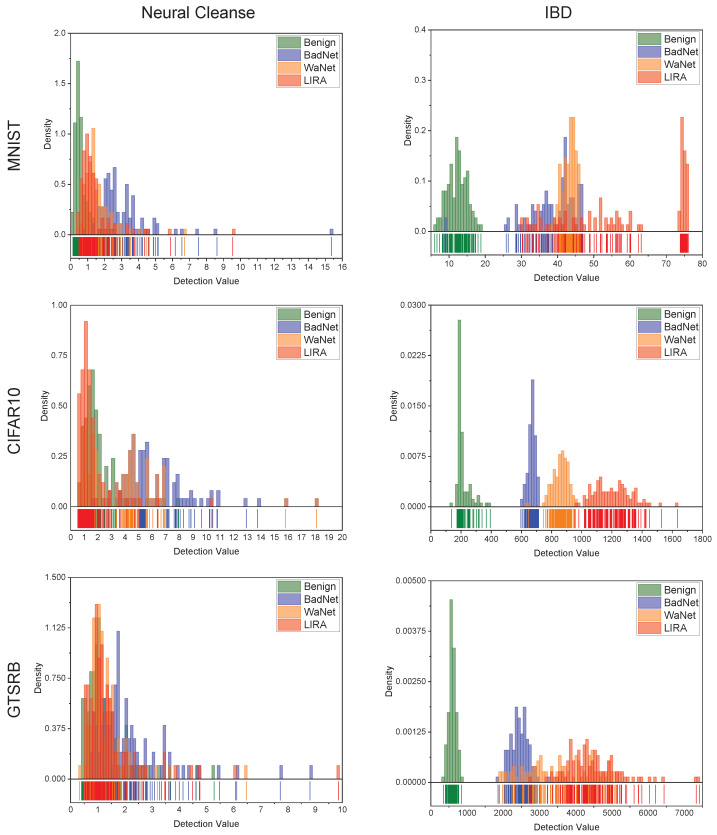
Histogram with RUG of IBD and Neural Cleanse when dealing with models trained on three different datasets. For each experiment, the test set contains 100 benign models and 3×100 poisoned models attacked by BadNet, WaNet, and LIRA, respectively.

**Table 1 sensors-22-08697-t001:** The evaluation result of our multivariate-interaction-based interpretable backdoor-detection method. The mean relationship values are calculated for benign and poisoned models using Equation (Equation 3).

Dataset	Attack	Mean Relationship Value	
		Target Class	Benign Classes
MNIST	Benign	-	32.7
	BadNet	85.7	33.2
	WaNet	114.8	32.9
	LIRA	122.5	33.5
CIFAR-10	Benign	-	48.6
	BadNet	108.3	46.5
	WaNet	182.4	50.2
	LIRA	212.9	49.5
GTSRB	Benign	-	43.7
	BadNet	93.8	45.8
	WaNet	149.2	44.7
	LIRA	196.5	46.0

**Table 2 sensors-22-08697-t002:** Performance comparison of IBD and Neural Cleanse against three different backdoor-attack methods on three datasets. The clean detection accuracy and the attack detection accuracy represent the mean detection accuracy of 100 benign models and 100 poisoned models, respectively. The detection accuracy of IBD and Neural Cleanse is calculated on 200 models where the false positive rate is set to be 0% ( i.e., all the clean models are classified correctly).

Dataset	Clean Accuracy	BadNet			WaNet			LIRA		
		Attack Success Rate	Neural Cleanse	IBD	Attack Success Rate	Neural Cleanse	IBD	Attack Success Rate	Neural Cleanse	IBD
MNIST	0.99	0.99	0.33	0.98	0.99	0.24	1	0.99	0.21	1
CIFAR10	0.84	0.98	0.25	1	0.98	0.17	1	0.98	0.16	1
GTSRB	0.99	0.98	0.27	1	0.99	0.18	1	0.99	0.15	1

**Table 3 sensors-22-08697-t003:** Time cost comparison of Neural Cleanse and IBD.

Dataset	Model	Time Cost (s)	
		Neural Cleanse	IBD
MNIST	VGG-like	180	0.1
CIFAR10	ResNet18	240	0.15
GTSRB	ResNet18	1800	0.48

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
