# Peer review of "IBD: An Interpretable Backdoor-Detection Method via Multivariate Interactions"

_sensors, 2022, doi:10.3390/s22228697_

Round 1
Reviewer 1 Report
Sensors-1940084-
In this paper, the authors proposed an interpretable backdoor detection method to against backdoor attacks. The multivariate interactions of different categorical features in benign and poisoned models were analyzed. Different categories were illustrated and the experimental results demonstrate the effectiveness and efficiency of this IBD.
It provided useful information. For the field of sensor and measurement, data collection and transmission have become important tropic to ensure the correction of data and reduce the invasion. Please provide some literature about the problems of data collection and transmission. Then illustrate the application of the technique introduced in this paper to help the fields of sensors.
Author Response
Please provide some literature about the problems of data collection and transmission. Then illustrate the application of the technique introduced in this paper to help the fields of sensors.
Thank you for your constructive comments. We add some literature to reveal how backdoor attacks influence sensors’ security as follows:
More and more sensors are deployed with deep learning models. However, existing studies on backdoor attacks against DNNs have demonstrated that backdoor attacks may occur throughout the lifetime of a deep learning model including model training[6,8,9], model transmission[7], and model application[10], which makes it a great threat against deep-learning-based sensors. Meanwhile, attackers may maliciously use the sensors and inject a backdoor[11]. Therefore, mitigating the threat of backdoor attacks is critical.

Reviewer 2 Report
The paper provides a new backdoor detection method that uses multivariate interactions. The paper is interesting and generally well written. However, some issues need to be solved:
1. The “interpretability” of results provided by the proposed method needs to be better explained. Moreover, presenting 1-2 examples in the experimental section (section 4) may help the reader to better understand this concept and how the proposed method deals with it.
2. “transferability” (lines 240-256) needs some explanations. The conclusion formulated in the second sentence (“Therefore, our proposed…”) is not entirely sustained by the previous sentence (“According to the…”);
3. The first keyword (DNN) needs to be replaced by “deep neural network”. Also third keyword (“interpretable”) is confusing and needs to be replaced;
4. English needs some polishing. For example, in line 86 instead of “Tang et al. [7] propose” it is written “Tang et al. [7] proposes” and this type of mistake may be found in many places inside the manuscript. There are also some typos, e.g. line 55 or 253.
Author Response
- The “interpretability” of results provided by the proposed method needs to be better explained. Moreover, presenting 1-2 examples in the experimental section (section 4) may help the reader to better understand this concept and how the proposed method deals with it.
Thank you for your constructive comments. To better explain and evaluate the interpretability, we add a subsection in the experiment section and provide a new table containing the evaluation results. The analysis is as follows:
First, we evaluate our Shapley-value-based interpretable method on three different datasets against three attack methods. Following the theoretical analysis of multivariate interactions. We calculate the mean relationship values of benign and poisoned models using Equation 3. According to Table 1, for benign classes, the mean relationship values of benign and poisoned models are similar, while the mean relationship values of the target classes are much larger. The experimental result demonstrates that features of different categories show an abnormally cooperative relationship in the poisoned models, which can be used to detect backdoor models.
- “transferability” (lines 240-256) needs some explanations. The conclusion formulated in the second sentence (“Therefore, our proposed…”) is not entirely sustained by the previous sentence (“According to the…”);
Thank you for your comments. We further explain why existing methods are not transferable and reconstruct the mentioned sentences as follows to better illustrate the transferability of IBD:
Most existing backdoor detection methods have some assumptions about the potential attack methods (e.g. Neural Cleanse[12] requires the trigger to be smaller than the natural features). Attackers can change the form of triggers guided by these assumptions, which makes existing detection methods not transferable to new attacks. IBD does not require defenders to have prior knowledge about the potential attacks. Therefore, it is transferable to detect different types of backdoor attacks including some state-of-the-art attack methods which can bypass most existing defense methods[9]. Meanwhile, it is difficult for attackers to manipulate the multivariate interactions of features in the target model, thus our methods are still robust against defense-known attacks.
- The first keyword (DNN) needs to be replaced by “deep neural network”. Also third keyword (“interpretable”) is confusing and needs to be replaced;
We rewrite the keywords as “deep neural network”, “backdoor detection”, and “interpretable deep learning” in our revised version.
- English needs some polishing. For example, in line 86 instead of “Tang et al. [7] propose” it is written “Tang et al. [7] proposes” and this type of mistake may be found in many places inside the manuscript. There are also some typos, e.g. line 55 or 253
We apologize for the misleading caused by the writing mistakes, and we have carefully checked the writing and corrected the errors (e.g., line 20 backdoor → backdoors, line 77 methods → method, line 88 transformission → transformation).

Reviewer 3 Report
Draw a diagram for the overall structure of view of your work. It will increase the visibility of your work.
Results and discussion section is not satisfactory. Even the claimed results are not reflected properly and compared with the previous method. Add a strong justification of your results.
You have not discussed the details of abbreviations such C1, C2, and C3 etc. Readers may take it wrong or get confused from this. Add abbreviation details where you have used short forms of any word.
How you are evaluating your theoretical analysis? Further, many of the claims in this paper are vague and just have text base justification. What is the contribution of this work?
The conclusion needs to revise. It looks too short and also include future directions for the readers at the end of this section.
Furthermore, the conclusion is not aligned with the abstract. Reconsider both
Some of the references are too old. Replace them with state of the art and recent studies. Even you have cited various survey articles. 1,2 are enough, remove the others.
I found many grammatical mistakes, typos, spelling mistakes in the article. The article needs extensive proofing and spell checks.
Round 2
Reviewer 2 Report
The authors have successfully solved my comments and concerns.
Author Response
Thank you for your efforts and constructive comments.
Reviewer 3 Report
I appreciate authors on their efforts towards revision and comments. But still there are various errors that must be addressed before publication.
1. "More and more sensors are deployed with deep learning models." Change this sentence to a meaningful sentence.
Similarly, there are few places left that need to be addressed with respect to grammar and proof reading.
As you are showing various results then you should write evaluation results and same symmetry should be followed throughout your manuscript.
2. You have removed some of the suggested survey references but the citations are not enough and some of the old references are still there that can be replaced with state of the art and recent articles from reputed venues. I suggest to remove preprints articles. You can consider these citations for your article or cite some others from 2021-2022:
i. Intrusion detection framework for the internet of things using a dense random neural network
ii. Attention Autoencoder for Generative Latent Representational Learning in Anomaly Detection
iii. Small-scale and occluded pedestrian detection using multi mapping feature extraction function and Modified Soft-NMS.
3. Accuracy is a vague term. If you are showing your results w.r.t to accuracy then you should mention the further details. Such as error accuracy, detection accuracy, performance accuracy etc.
Please address this issue in the whole manuscript.
Author Response
Please refer to the pdf file.
